# Dynamin-Related Proteins Enhance Tomato Immunity by Mediating Pattern Recognition Receptor Trafficking

**DOI:** 10.3390/membranes12080760

**Published:** 2022-08-01

**Authors:** Meirav Leibman-Markus, Silvia Schuster, Beatriz Vasquez-Soto, Maya Bar, Adi Avni, Lorena Pizarro

**Affiliations:** 1School of Plant Sciences and Food Security, Faculty of Life Sciences, Tel Aviv University, Tel Aviv 6997801, Israel; meiravleibman@gmail.com (M.L.-M.); schus@tauex.tau.ac.il (S.S.); lpavni@tauex.tau.ac.il (A.A.); 2Department of Plant Pathology and Weed Research, Institute of Plant Protection, ARO, Volcani Institute, Rishon LeZion 7505101, Israel; mayabar@volcani.agri.gov.il; 3Institute of Agri-Food, Animal and Environmental Sciences, Universidad de O’Higgins, Rancagua 2820000, Chile; bgvasquez@uc.cl

**Keywords:** dynamin-related protein, DRP1, endomembrane trafficking, FLS2, LeEIX2, tomato, defense responses

## Abstract

Pattern recognition receptor (PRR) trafficking to the plasma membrane and endocytosis plays a crucial role in pattern triggered immunity (PTI). Dynamin-related proteins (DRPs) participate in endocytosis and recycling. In Arabidopsis, DRP1 and DRP2 are involved in plasma membrane scission during endocytosis. They are required for the PRR FLS2 endocytosis induction and PTI activation after elicitation with flg22, the MAMP recognized by FLS2. In tomato, SlDRP2A regulates the PRR LeEIX2 endocytosis and PTI activation in response to EIX, the MAMP recognized by LeEIX2. However, it is unknown if other DRPs participate in these processes. Taking advantage of bioinformatics tools, we selected SlDRP2B among the eight DRP2 tomato orthologues to study its functionality in trafficking and plant immunity. Through transient expression of SlDRP1B and its dominant-negative mutant on *Nicotiana benthamiana* and *Nicotiana tabacum,* we analyzed SlDRP1B function. We observed that SlDRP1B is physically associated with the LeEIX2 and modifies LeEIX2 trafficking, increasing its presence in endosomes. An enhancement of EIX-elicitated defense responses accompanies the role of SlDRP1B on LeEIX endocytosis. In addition, SlDRP1B overexpression enhanced flg22-elicited defense response. With these results, we conclude that SlDRP1B regulates PRR trafficking and, therefore, plant immunity, similarly to the SlDRP2A role.

## 1. Introduction

During pathogen colonization, plants sense an intruder by recognizing exogenous molecules through the plasma membrane (PM) and intracellular receptors. Pattern recognition receptors (PRRs) are the PM receptors that recognize microbial-associated molecular patterns (MAMPs) [1]. PRRs are part of the first layer of plant immunity, sensing pathogens and triggering defense responses to halt pathogen invasion. There are two main groups of PRRs: receptor-like kinases (RLKs) and receptor-like proteins (RLPs). RLKs are generally composed of an extracellular recognizing domain, such as a leucine-rich repeat (LRR) domain thought to be involved in ligand binding; a transmembrane domain; and an intracellular kinase domain involved in signal transduction [2]. RLPs have an extracellular recognizing domain and a transmembrane domain, but lack any obvious domains in charge of intracellular signaling. RLPs are assumed to form receptor complexes with RLKs or receptor-like cytoplasmic kinases to transduce defense signals to target proteins [3,4,5]. PRRs are integral membrane proteins synthesized in the endoplasmic reticulum (ER), and carried to the PM through the Golgi, early endosomes/trans-Golgi network (EE/TGN). PRRs must be adequately delivered and localized at the PM for sensing MAMPs from the apoplast [6].

Roles for endocytosis and cellular trafficking in plant cellular responses to invading pathogens are well documented [6,7]. Receptor-mediated endocytosis (RME) serves as a mechanism for both signaling initiation and termination through the degradation of activated PRR complexes after internalization [7,8]. Plant cells require PRR presence at the PM to mount effective defenses. It has been suggested that, from an evolutionary perspective, targeting plant vesicular cellular trafficking is a strategy adopted by many pathogens through their effector molecules [9,10,11]. Plant endocytic trafficking components are diverted towards pathogen infection sites when required, reinforcing the importance of cellular trafficking in immune responses [12,13,14,15]. The LRR-RLK FLS2 recognizes the epitope flg22 from flagellin [16]. FLS2 undergoes constitutive endocytosis and recycling between the PM and the trans-Golgi network as a mechanism of maintaining its steady-state level at the PM [17,18]. Upon ligand stimulation, FLS2 transmits from the PM and localizes to endomembrane vesicles in a BAK1-dependent manner [18,19,20], and then traffics toward the vacuole for degradation [21]. FLS2 thus traffics through different endocytic pathways, aimed both at maintaining cellular readiness to sense pathogens and mounting an efficient response once exposed to an actual pathogen.

The LRR-RLP LeEIX2 traffics on endomembrane compartments. The binding of the fungal Xylanase-11 ligand (EIX) to LeEIX2 induces receptor-mediated endocytosis [22]. Inhibition of endosome formation reduces EIX-mediated responses, while arresting post internalization cellular trafficking increases these responses [23,24,25]. This suggests that LeEIX2 transmits defense signals from endosomes. Interestingly, although much of the defense signals and signal-propagation mechanisms differ between LeEIX2 and FLS2 [20,23], the involvement of the endomembrane system as a tool for both signal propagation and maintenance of immune system readiness is conserved. This testifies to the importance of cellular trafficking in plant response to attacking pathogens as a general mechanism. Indeed, it was recently reported that overexpression of PRRs is sufficient to induce localized immune responses [26], and furthermore, plants that are constitutively immuno- activated have increased expression of a variety of PRRs from both the RLK and RLP classes in steady-state [27,28].

Dynamin-related proteins (DRPs) are mechanochemical GTPases that remodel membranes in diverse cellular processes [29,30]. In plants, DRPs are classified into six families, and are involved in vesicle formation and scission during endocytosis and post-Golgi traffic [7,31,32]. They are also critical for homotypic membrane fission and fusion in different organelles [31,32]. DRP1 and DRP2 have been implicated in PM scission during endocytosis [33,34]. In plant immunity, AtDRP1 and AtDRP2 are required for flg-22-induced FLS2 endocytosis, and are involved in PTI in response to flg22 and *Pseudomonas syringe* in *Arabidopsis thaliana* [25,35,36]. As such, DRP1s and DRP2s are involved in pathogen resistance; they are targeted by pathogen effectors in several cases [34,35,36,37,38,39]. We previously reported that DRP2A of tomato associates with LeEIX2, regulating its trafficking, defense signaling, and defense response [25]. As both AtDRP1 and AtDRP2 are required for endocytosis of PRRs, we aim to elucidate if the same occurs in tomato. Tomato has eight orthologs in the AtDRP1 sub-family [25,40]. Here, we identified a drp1 family member protein based on homology and characterized its role in PRR endocytosis and plant immunity.

## 2. Materials and Methods

### 2.1. Plant Materials and Growth Conditions

*Nicotiana tabacum* cv samsun NN, *N. benthamiana,* and *S. lycopersicum* cv M82 were grown from seeds in soil (Green Mix; Even-Ari, Ashdod, Israel) in a growth chamber, under long-day conditions (16 h:8 h, light:dark) at 24 °C.

### 2.2. Bioinformatic Analysis

Protein domain analysis of tomato DRP1 orthologues [25] was performed using HMMER: biosequence analysis using profile hidden Markov models [41], and a domain representation was drawn. Expression analysis of each of the DRP1 orthologues, and co-expression analysis of SlDRP1A, SlDRP1B, and SlDRP2A, were made using Genevestigator® software (all samples database, Zurich, Switzerland) [42]. A protein–protein interacting (PPI) network was built using the physical PPI analysis with the PTIR database [43] and plotted using the Cytoscape program.

Gene ontology overrepresentation analysis was performed by PANTHER GO overrepresentation test. A set of 180 out of 191 SIDRP1B co-expressed genes were found and analyzed, using the complete *S. lycopersicum* genome as a reference list (34,652 genes), and Fisher’s test and false discovery rate (FDR) as statistical methods [44]. Data visualization was performed using the Python library plotline (based on ggplot2) and considering fold-enrichment, overrepresentation significance (*p*-value), and number of genes related to each GO term (count). For corroborating the expression level of Solyc01g095970, Solyc05g050600, Solyc01g005310, and Solyc08g077360, specific real-time PCR primers were designed (Appendix A). RNA was extracted from M82 tomato leaves (2 months old). M-MLV reverse transcriptase (Promega, Madison, WI, USA) and oligo-d(T) primers were used for cDNA synthesis according to manufacturer’s instructions. qRT-PCR was performed according to the Fast SYBR qPCR Master Mix (BioGate, Yad HaShmona, Israel), using a Rotor-Gene Q machine (Qiagen, Valencia, CA, USA). Relative expression was determined using the following formula: Relative expression = 2^Ct Hkp–Ct SlDRP1X^. For determining relative expression, we used RPL8 (Solyc10g006580) housekeeping gene [45,46]. Three RNA biological replicates were used; each reaction was performed in triplicate.

### 2.3. Transient Expression 

Binary vector clones were introduced by electroporation into *Agrobacterium tumefaciens* strain GV3101. *Agrobacterium* cells were grown in LB medium containing 50 mg/L Kanamycin, 40 mg/L Gentamycin, and 100 mg/L Rifampicin overnight at 28 OC, diluted into VIR induction medium (50 mM MES pH 5.6, 0.5% (*w*/*v*) glucose, 1.7 mM NaH_2_PO_4_, 20 mM NH_4_Cl, 1.2 mM MgSO_4_, 2 mM KCl, 17 μM FeSO_4_, 70 μM CaCl_2_, and 200 μM acetosyringone), and grown for 6 additional hours until OD600 reached 0.4–0.6. Suspensions containing single or mixed *Agrobacterium* cultures were diluted to a final OD600 of 0.15–0.2 in VIR induction medium. Cultures were infiltrated abaxially with a needless syringe into leaves of *N. tabacum* cv samsun NN or *N. benthamiana*. Leaves were harvested 40 h after injection for ethylene measurement, ROS measurement, CoIP, BiFC, or confocal microscopy analysis.

### 2.4. Cloning of SlDRP1B and Site-Directed Mutagenesis

SlDRP1B cDNA C-terminally tagged with GFP or mCherry were amplified from *S. lycopersicum* cv M82 and cloned into the SalI, XbaI sites of pBINPLUS, following the strategy used by Pizarro et al., 2019 [25] using the following primers: SlDRP1B forward primer 5′-CCGTCGACATGGGAATCTCATAGCATTAG-3′ and SlDRP1B reverse primer 5′-CTCTAGACTTAGACCATGCTACTGAATC-3′. Amplified fragments were cloned between the CAM35SΩ promoter containing the Ω translation enhancer signal and the Nos terminator.

For site-directed mutagenesis, GTPase domain and the catalytic site were predicted using Conserved Domain search from National Center of Biotechnology Information website2, following the strategy used by Pizarro et al, 2019 [25]. Site-directed mutagenesis on the GTPase catalytic site of SlDRP1B was created using the Q5 site-directed mutagenesis kit (NEB #E0554) and the following primers: forward 5′-RTGTCCACCGACAACAGCGU-3′ and reverse 5′-FGAGTTCTGGGGCATCTTCGGTGCTTG-3′.

### 2.5. Co-Immunoprecipitation

Co-immunoprecipitation assays were performed as described by Leibman-Markus [47]. *N. benthamiana* leaves transiently co-expressing LeEIX2-HA and SlDRP1B-GFP were harvested 40 h after infiltration. Leaf petioles were immersed in EIX 3 μg/mL (or water as mock) for seven minutes and then transferred to the water for an additional seven minutes. A total of 500 mg leaf tissue was used for co-immunoprecipitation, with 13 μL α-HA Affinity Matrix (Roche, Indianapolis, ID, USA). Samples were run in SDS-PAGE, blotted onto nitrocellulose membranes, and incubated with antibodies as required: rat α-GFP (Chromotek, Planegg-Martinsried, Germany) and mouse α-HA (Biolegend, San Diego, CA, USA).

### 2.6. BiFluorescence Complementation

LeEIX2 cytoplasmic domain and SlDRP1B cDNA were cloned into the Spe1 site of pSY751 and pSY752, downstream of the N-terminal fragment of YFP (YN) and the C-terminal fragment (YC), respectively, using the following primers: LeEIX2 forward primer 5′-GGGGCCTTTTAGGCTG-3′ and LeEIX2 reverse primer 5′-CTGGCGGCCGCTCAGTTCCTTAGCTTTCCC-3′; SlDRP1B forward primer 5′-CCACTAGTATGGAGAATCTCATAGCATTAG-3′ and SlDRP1B reverse primer 5′-CCACTAGTTTACTTAGACCATGCTACTG-3′. The resulting plasmids were used for transient expression assay in *N. benthamiana* leaves.

### 2.7. ROS Burst Assay

ROS burst was measured as previously described by Leibman-Markus [47]. Leaf disks (0.5 cm diameter) were taken from tobacco plants transiently expressing SlDRP1B tagged with GFP, mCherry, or HA, and free GFP was used as control. Disks were floated in 250 μL ddH_2_O in a white 96-well plate for 4–6 h at room temperature. After incubation, water was removed entirely and ROS measurement reaction containing EIX 1 μg/mL or Flg22 1 mM was added, and light emission was immediately measured using a micro-plate luminometer (Turnerbiosystems Veritas, Santa Clara, CA, USA).

### 2.8. Ethylene Production Assay

Ethylene biosynthesis was measured as previously described by Leibman-Markus [47]. Leaf disks (0.9 cm diameter) were taken from transiently expressing tobacco plants. Five disks were sealed in each 10 mL flask containing 1 mL assay medium (with or without 1 μg/mL EIX) and incubated with shaking for four hours at room temperature. Ethylene production was measured by gas chromatography (Varian 3350, Varian, Palo Alto, CA, USA).

### 2.9. Live-Cell Imaging

Confocal microscopy images were acquired using a Zeiss LSM780 confocal microscope system using Objective C-Apochromat 40x/1.2 W Corr M27 (Zeiss, Oberkochen, Germany). Microscopy configuration includes two tracks: Track 1 for collecting chlorophyll autofluorescence using, excitation laser wavelength of 633 nm (2% power), and emission collection range of 652–721 nm; Track 2 for collecting GFP and dsRed/mCherry fluorescence using excitation laser of 488 nm (5% power) and 561 nm (2% power), respectively. GFP and dsRed/mCherry emission was collected in the range of 493–535 nm and 588–641 nm, respectively. For YFP image acquisition, an excitation laser of 514 nm (5% power) was used, and emission was collected in the 522–530 nm range. 1024X1024 pixels images were acquired using a pixel dwell time of 1.27, pixel averaging of 4, and pinhole of 1 airy unit.

### 2.10. Confocal Image Analysis

Image analysis was conducted using Fiji-ImageJ on the original files [48]. The analysis was performed on a single epidermal cell region of interest (ROI) where both proteins expressed with similar intensity. Colocalization of SlDRP1B and the endosome markers was determined using Coloc2 function. The 3D object counter function was used for quantifying compartment density and size. Fluorescence ratio was performed by quantifying the integrated pixel intensity in the PM divided by the integrated pixel intensity in the whole cell. 

## 3. Results

### 3.1. Searching for a Functional Tomato DRP1 Involved in PRR Endocytic Trafficking

DRP1 proteins contain three dynamin core domains: an N-terminal GTPase domain, a middle domain, and a GTPase effector domain (GED) [41,42,43,44]. To identify a functional tomato DRP1, we searched for the DRP1 orthologs in the tomato genome, filtering in by homology to AtDRP1A (At5g42080) and AtDRP1B (At3g61760), using the Sol Genomics Network [47]. Then, we analyzed the putative orthologues using three available bioinformatics platforms: HMMR EMBL-EBI, GENEVESTIGATOR, and PTIR [41,42,43,44]. We found eight DRP1s gene orthologues to DRP1s; among them, four possessed the three canonical domains of a DRP1: Solyc01g095970, Solyc05g050600, Solyc01g005310, and Solyc08g077360 (Figure 1A, Appendix A). Expression levels of the DRP1 orthologues were analyzed using GENEVESTIGATOR® bioinformatic software. When examining a dataset of 877 samples, Solyc01g095970, Solyc05g050600, and Solyc01g005310 have the highest average expression among the eight DRP1 orthologues (Figure 1B). Expression analysis of Solyc01g095970, Solyc05g050600, Solyc01g005310, and Solyc08g077360 was corroborated via real-time PCR using RNA from tomato leaves (Appendix A). 

To collect information about the possible interactors of these genes, we used the PTIR database. This analysis predicted interactors for three tomato putative DRP1 proteins: Solyc01g095970 (SlDRP1A), Solyc05g050600 (SlDRP1B), and Solyc01g005310 (SlDRP1C). SlDRP1A and SlDRP1B were predicted to share the same interactors, while SlDRP1C was found to differ in some putative interactors (Figure 1C, Appendix A). Interestingly, all of them interact with SlDRP2A (Figure 1C, Appendix A). In addition, SlDRP1A and B share four out of five interactors with SlDRP2A (Figure 1C, Appendix A). The interaction between DRP1 and DRP2 during the formation of clathrin-coated vesicles was previously described in Arabidopsis [35]. We searched for the genes that share similar expression profiles with SlDRP1A, SlDRP1B, and SlDRP1C. We set a cut-off = 0.8 in Pearson’s correlation factor for co-expression. The highest Pearson’s correlation value obtained for SlDRP1A was 0.74; therefore, under our criterion, SlDRP1A does not co-express with any gene (Appendix A). SlDRP1B was found to be co-expressed with 191 genes, including SlDRP2A (Appendix A). Interestingly, 109 of the genes found to be co-expressed with SlDRP1B were also co-expressed with SlDRP2A (Figure 1C, Appendix A). The list of genes correlatively expressed with both SlDRP1B and SlDRP2A (Appendix A) includes genes such as two Coatomers (Solyc04g080980 and Solyc12g043000), a DRP3 (Solyc11g043210), a Myosin (Solyc07g041150), and a Clathrin Heavy Chain (Solyc05g052510). These results suggest that SlDRP1B could function in the processes described for SlDRP2A. In contrast, SlDRP1C was found to have 382 co-expressed genes; however, none of them were co-expressed with SlDRP2A.

Additionally, gene ontology enrichment analysis for molecular function classification of SlDRP1B co-expressed genes showed overrepresentation of genes belonging to the classification terms “Vesicle-mediated transport”, “Localization”, “Transport establishment of localization”, and “Cellular localization” (Appendix A). Given the results of the bioinformatic analysis, we determined that SlDRP1B showed the most promise to be a functional DRP1 family protein and proceeded to characterize the roles of SlDRP1B in trafficking and defense.

### 3.2. SlDRP1B Subcellular Localization

To characterize SlDRP1B and determine its subcellular localization, we conducted confocal microscopy experiments with fluorescently labeled SlDRP1B. Confocal images of *N. benthamiana* epidermal cells reveal that SlDRP1B has a dual localization (Figure 2), as reported for Arabidopsis members of this sub-family [33,49]. SlDRP1B forms discrete foci at the PM (marked in white circles), ranging in diameter between 200–550 nm (Figure 2A,B) [33,50], and colocalizing with a PM marker such as Flot1 (Pearson correlation coefficient of 0.83) [51]. This result suggests that SlDRP1B, similarly to AtDRP1, is involved in the formation of endocytic vesicles at the PM [33]. In addition, SlDRP1B also appears at punctuated structures (marked with white arrowheads), resembling endosomes (Figure 2A). Comparing the subcellular localization of SlDRP1B and SlDRP2A, we observed that SlDRP2A distribution is enriched at the PM (Appendix A) [25].

To better examine the subcellular localization of SlDRP1B and its punctuated pattern, we expressed SlDRP1B together with several endomembrane markers. SlDRP1B c-localizes with late endosomes/multivesicular body (LE/MVB) marker, RabG3f [52,53], having a Pearson correlation coefficient of 0.69 (Figure 2A,B). No colocalization was observed when co-expressing SlDRP1B with the EE/TGN marker, RabA1e [54,55,56], or with the Golgi marker, ST [56,57], having Pearson correlation coefficients of 0.54 and 0.22, respectively (Figure 2A,B). Surprisingly, although it colocalized with RabG3f, SlDRP1B did not colocalize with the LE/MVB/PVC marker FYVE [54], having a negative Pearson correlation coefficient of-0.04 (Figure 2A,B). This suggests that SlDRP1B localizes to a sub-population of LE/MVB. Interestingly, RabG3f affects SlDRP1B sub-cellular distribution, causing it to shift towards punctuated compartments, leading to an increase in SlDRP1B compartment number and size (marked with yellow arrowheads, Figure 2A,C,D). These changes might be a result of the RabG3f function or due to its overexpression.

### 3.3. SlDRP1B Associates with LeEIX2 and Enhances Defense Responses Mediated by EIX

We previously observed that SlDRP2A enhances defense responses mediated by LeEIX2 and FLS2 [25]. In order to examine a possible physical association between LeEIX2 and the tomato SlDRP1B ortholog, we performed co-immunoprecipitation (CoIP) and bi-fluorescence complementation (BiFC) experiments. We transiently co-expressed tomato SlDRP1B-GFP and LeEIX2-HA in *N. benthamiana* (an EIX nonresponsive plant species), and pulled down LeEIX2-HA using HA affinity beads. SlDRP1B was successfully pulled down in the presence of LeEIX2. The association between the two proteins was dramatically enhanced when pretreated with the ligand EIX (Figure 3A). We performed BiFC analysis to further validate the association. When co-expressing YN-LeEIX2 (fused to the N-terminal portion of YFP) and YC-SlDRP1B (fused to the C-terminal portion of YFP), we observed reconstitution of YFP fluorescence, indicating protein association (Figure 3B).

To examine the effect of SlDRP1B overexpression on ROS burst and ethylene production elicited by EIX, we transiently expressed SlDRP1B in *N. tabacum* cv Samsun NN (EIX responsive cultivar) followed by EIX application. SlDRP1B enhanced LeEIX2-mediated defense responses compared to control, leading to an increase in ROS burst and ethylene production (Figure 4). Based on bioinformatic analysis, we mutated the GTPase active site (Lysine 47 to Alanine, SlDRP1BK47A), creating a predicted loss of function. The mutated form lost the ability to enhance EIX-mediated defense responses, effecting a decrease in EIX-dependent ROS burst (Figure 4).

We next tested the possibility of SlDRP1B affecting the signal transduction cascades of additional PRR proteins. We turned to the hallmark PTI receptor FLS2, and examined a possible role of SlDRP1B in flagellin-mediated defense responses. Overexpression of SlDRP1B enhanced ROS burst upon flg22 elicitation by more than 60% (Appendix A), suggesting that SlDRP1B’s role as a positive immunity regulator is not restricted to LeEIX2-mediated defense responses.

In our bioinformatics analyses (Figure 1), SlDRP1B and SlDRP2A were also likely co-expressed with a DRP3 family protein, Solyc11g043210 (Appendix A), here named SlDRP3A. We cloned SlDRP3A and conducted similar experiments to those detailed here for SlDRP1B, finding that SlDRP3A also enhances LeEIX2-mediated defense (Appendix A).

### 3.4. SlDRP1B Affects LeEIX2 Endosomal Distribution

DRPs are essential modulators of endocytosis, and endocytosis affects LeEIX2 signaling [23,25,31]; therefore, we studied the effect of SlDRP1B on LeEIX2 subcellular localization. As previously published, LeEIX2 endosomes increase in both size and density upon EIX treatment (Figure 5), reflecting endocytosis involvement in EIX-induced signaling [24,25]. When overexpressing SlDRP1B, we observed a threefold increase in LeEIX2 endosome size and density in basal conditions (without EIX elicitation, Figure 5). As expected, SlDRP1B_K47A_ mutant the loss of function of SlDRP1A, behaves similarly to the control, and does not exhibit any effect on LeEIX2 endosomes (Figure 5). When overexpressing SlDRP1B, EIX elicitation led to a mild increase in LeEIX2 endosome size and density, similar to both the basal control and the control under elicited conditions (Figure 5). When comparing LeEIX2 endosome size and density in both basal and ligand-elicited conditions in the background of SlDRP1B overexpression, we observed a decrease in LeEIX2 endosome size and density, rather than an increase. In samples overexpressing the mutated SlDRP1B_K47A_, we see an increase in LeEIX2 endosome size upon elicitation, but a decrease in LeEIX2 endosome density (Figure 5). Interestingly, LeEIX2 fluorescence at the PM does not change significantly in any of the conditions analyzed (Appendix A), except for SlDRP1B. When SlDRP1B is overexpressed, LeEIX2 PM fluorescence is lower than in the control. However, after elicitation, LeEIX2 level at the PM increases, suggesting a possible activation of a recycling pathway. 

## 4. Discussion

PRRs are PM resident proteins. At the PM, they are available to detect extracellular elicitors. PRRs undergo a constant flux of endocytosis; from endosomes, they can be sorted back to PM or towards the vacuole. The fate of the PRR-trafficking directly impacts immune signaling; recycling to the PM enables PRR availability for the perception of newly arriving elicitors/MAMPs, while trafficking towards vacuole leads to PRR degradation, reducing PRR availability at the PM, resulting in signal attenuation [6,8]. Using various chemical modulators that affect different stages of endomembrane trafficking demonstrated that LeEIX2 signals from within early endosomes [23]. Upon EIX elicitation, we see an increase in both size and density of LeEIX2 endosomes (Figure 5) [24,25]. Interestingly, in previous work, we observed that overexpression of Prenylated Rab acceptor1 (PRA1) in tomato results in increased vacuolar degradation of PRRs and attenuated levels of PRRs at the PM, coupled with reduced defense signaling [57]. This and other works have demonstrated that PRR levels at the PM directly impact the amplitude of defense responses mounted by the plant cell. Furthermore, in another work, we found that overexpression of PRA1 leads to disease susceptibility [58], confirming that this attenuation in defense signaling caused by PRR insufficiency has “real” implications for plant health. Highlighting PRR relevance in plant defense, a functional interaction between PRR and pathogen effector intracellular receptors from the type NLR has been shown, indicating that PRRs are also components of effector-triggered immunity [28].

PRRs’ signal recognition occurs at the PM, where they form signaling complexes. Both LeEIX2 and FLS2 undergo constant recycling to the PM for further availability [18,59]. We previously reported the involvement of SlDRP2A in LeEIX2 and FLS2 signaling [25]. In that work, we observed significant changes to the size and cellular distribution of endomembrane compartments upon overexpression of SlDRP2A, as well as enhanced defense responses following both EIX and flg-22 elicitation. Here, we observed similar results for SlDRP1B, and enhanced defense also with SlDRP3A, indicating that multiple DRPs can have similar or complementary functions. In *A. thaliana,* it was proven that DRP1 and DRP2 work together in clathrin-dependent endocytosis, plant growth, and flg22-driven immune responses [33,34]. Remarkably, in tomato, SlDRP1B, SlDRP2A, and SlDRP3A are co-expressed, and overexpression of all three DRPs led to an increase of 50–100% in defense signaling output, depending on the parameter being measured (Figure 3 and Figure 4; Appendix A) [25]. This suggests that different DRPs from different families could have complementary functions in modulating PRR subcellular distribution. Notably, DRP3s are related to membrane fission and fusion of peroxisomes and mitochondria [60,61,62], possibly indicating that SlDRP3A’s effect on regulating PRR function could be indirect.

The increase in endomembrane compartment density and size, coupled with the increase in defense signaling, reinforces the idea that some of the PRR signals are emanating directly from endomembrane compartments and suggests that the altered endomembrane structures observed upon DRP overexpression do not hinder signal propagation, at least in the short-term. Interestingly, LeEIX2 endosomes are significantly bigger when SlDRP1B is overexpressed in basal conditions (without EIX elicitation). LeEIX2 endosomes return to their original size after EIX treatment (Figure 2A,C). These results suggest that SlDRP1B expression dramatically affects LeEIX2 endosomes, causing an early onset of LeEIX2 endocytosis induction, independently of EIX elicitation. EIX application seems to reverse the SlDRP1B effect partially, perhaps by facilitating the downstream trafficking of the LeEIX2 endosomes, or recycling to the PM, and thus releasing a possible bottleneck created by SlDRP1B overexpression. These results are similar to those previously observed for SlDRP2A [25]. While DRP2A and DRP1B share similar domains, DRP1B lacks the PH and PRM motifs present in DRP2A (Appendix A), suggesting that these motifs may not be required for the interactions of DRPs with PRRs, or for defense signaling enhancement. Both DRPs are localized to endomembrane compartments. DRP2A has a significant PM presence [25], while DRP1B is more specifically localized to a subset of late endosomes (Figure 2). This may suggest that these different DRPs interact with different PRRs in different compartments, or with PRRs that are in a different “signaling state”, possibly providing specificity needed to discern between “activated” receptors and “recycling-bound” receptors that are targeted either for degradation or for return to the PM, though further work is needed to elucidate this point.

Here, we show the involvement of DRPs from different families in PRR-mediated defense responses. DRP1B (Figure 3, Appendix A) and DRP2A [25] both enhanced PRR-mediated defense signaling of the RLP LeEIX2 and the RLK FLS2, demonstrating the generality of DRP-mediated cellular processes in PRR function. Overexpression of DRPs from different families [25] (Figure 3 and Figure 4) resulted in alterations to cellular signaling and PRR trafficking, likely underlying the observed increases in defense output and adding to the evidence of the roles of cellular trafficking in plant immune signal propagation. The similar function of the different DRPs in defense, coupled with their slightly different subcellular localization and protein domains, suggests that they may have similar functions in different cellular locales, or similar functions within complexes with different proteins (Figure 1C). Future work will tease out the different specificities of different DRPs, and elucidate general mechanisms underlying the functionalities of dynamin-family proteins in different immunity and pathogenicity contexts.

## Figures and Tables

**Figure 1 membranes-12-00760-f001:**
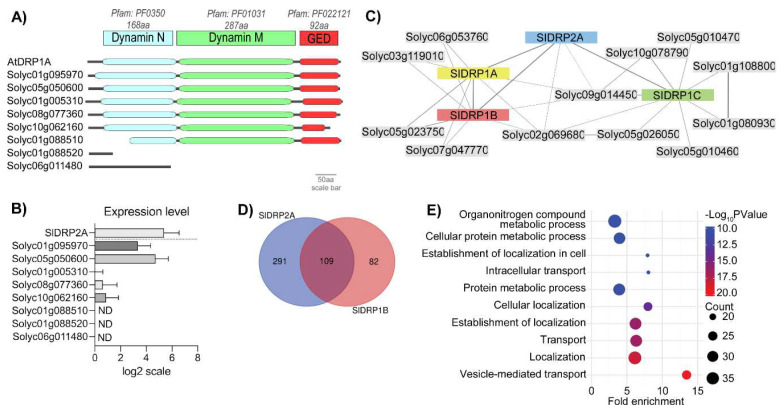
Bioinformatic analysis of tomato DRP1 orthologues. (**A**) Protein domain analysis was per this figure. Orthologues: Solyc01g095970, Solyc05g050600, Solyc01g005310, Solyc08g077360, Solyc10g062160, Solyc01g088510, Solyc01g088520, and Solyc06g011480. (**B**) Gene expression analysis of tomato DRP1 orthologues was performed using GENEVETIGATOR®. (**C**). Interactome network of SlDRP1A, SlDRP1B, and SlDRP1C based on PTIR database (Appendix A). (**D**) Venn diagram of SlDRP1B and SlDRP2A co-expressed genes based on GENEVETIGATOR® database. (**E)** GO overrepresentation analysis of SlDRP1B co-expressed genes.

**Figure 2 membranes-12-00760-f002:**
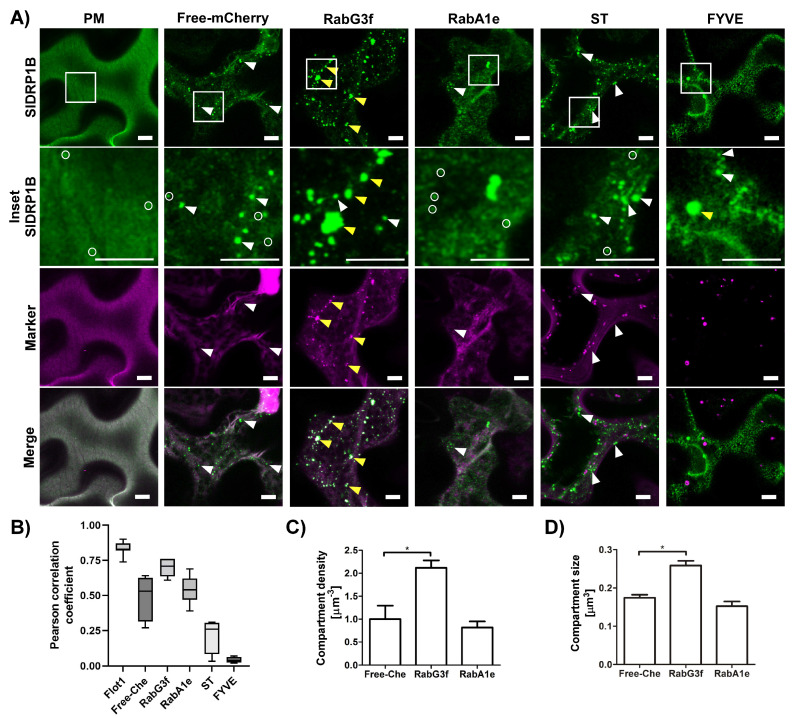
SlDRP1B localizes to a subset of MVB/LE/PVC compartments. Localization of SlDRP1B was analyzed by confocal microscopy of *N. benthamiana* epidermal cells transiently expressing SlDRP1B-GFP and different endomembrane compartment markers as indicated. Images were acquired using a Zeiss LSM 780 microscope. (**A**) Representative images of SlDRP1B-GFP colocalization with Flot1-mCherry, free mCherry, RabG3f-mCherry, RabA1e-mCherry, ST-mCherry, and FYVE-dsRED are shown. White and yellow arrowheads point to SlDRP1B compartments and SlDRP1B compartments colocalizing with the markers, respectively. Circles indicate SlDRP1B foci. Squares indicate the inset region. Scale bar 7.5 µm. (**B**) Colocalization analysis between SlDRP1B and the markers. (**C**,**D**) SlDRP1B endosomes density Endosome density was normalized to control (Free mCherry). Control = 0.013 endosomes per µm^2^. (**B**–**D**) Twelve images each were analyzed. (**C**,**D**). Error bars represent mean ± SEM. Asterisks indicate significant differences with free mCherry control (*t*-test, * *p* < 0.05).

**Figure 3 membranes-12-00760-f003:**
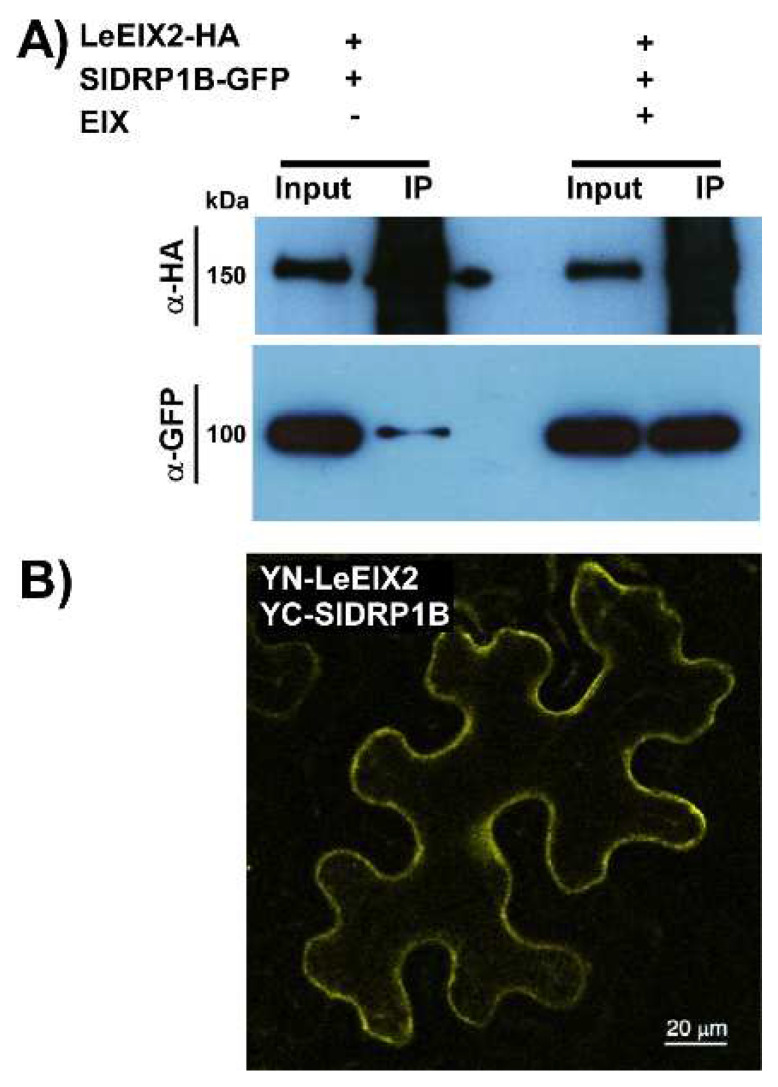
SlDRP1B associates with LeEIX2. (**A**) *N. benthamiana* was transiently transformed with LeEIX2-HA and SlDRP1B-GFP. Leaves were harvested and treated with EIX or water (mock). Input and immunoprecipitated samples were subjected to SDS-PAGE and immunoblot with anti-HA antibodies to detect LeEIX2-HA, and anti-GFP antibodies to detect SlDRP1B-GFP. (**B**) BiFC visualization of the association between LeEIX2 and SlDRP1B. *N. benthamiana* leaves transiently expressing YN-LeEIX2 cytoplasmatic domain and YC-SlDRP1B as indicated. Leaf sections were visualized using a laser-scanning-meta confocal microscope. Scale bar 20 μm.

**Figure 4 membranes-12-00760-f004:**
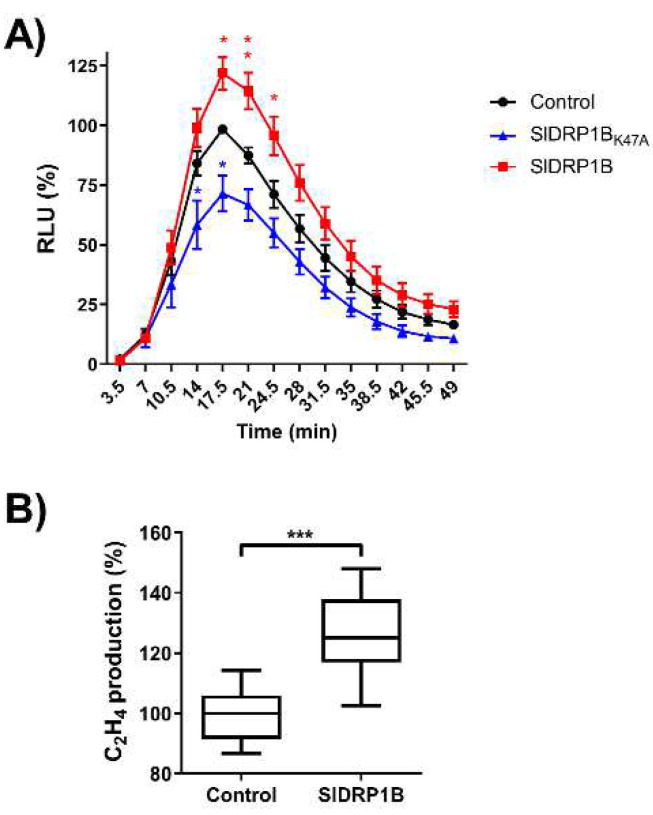
Effect of overexpression of SlDRP1B on LeEIX2-mediated defense responses. Leaf disks of *N. tabacum* transiently expressing SlDRP1B-tagged, SlDRP1B_K47A_-tagged, or free-tag (control) were harvested 48 h after transformation. (**A**) Luminescence (RLU) was measured immediately after EIX application. Error bars represent the average +SEM of 7 independent experiments, N = 12. The average value of control peak was defined as 100%. Time points represent the average ± SEM. Asterisks indicate significant differences with control treatment (two-way ANOVA, * *p* < 0.05, ** *p* < 0.01). (**B**) Ethylene (C_2_H_4_) biosynthesis was measured 4 h after EIX application. Average value of control was defined as 100%. Boxplots represent minimum to maximum values, with boxes representing the inner quartile ranges, whiskers representing the outer quartile ranges, and the line in the box representing the median, of 3 independent experiments, with asterisks denoting significant differences to control treatment (*t*-test, *** *p* < 0.001).

**Figure 5 membranes-12-00760-f005:**
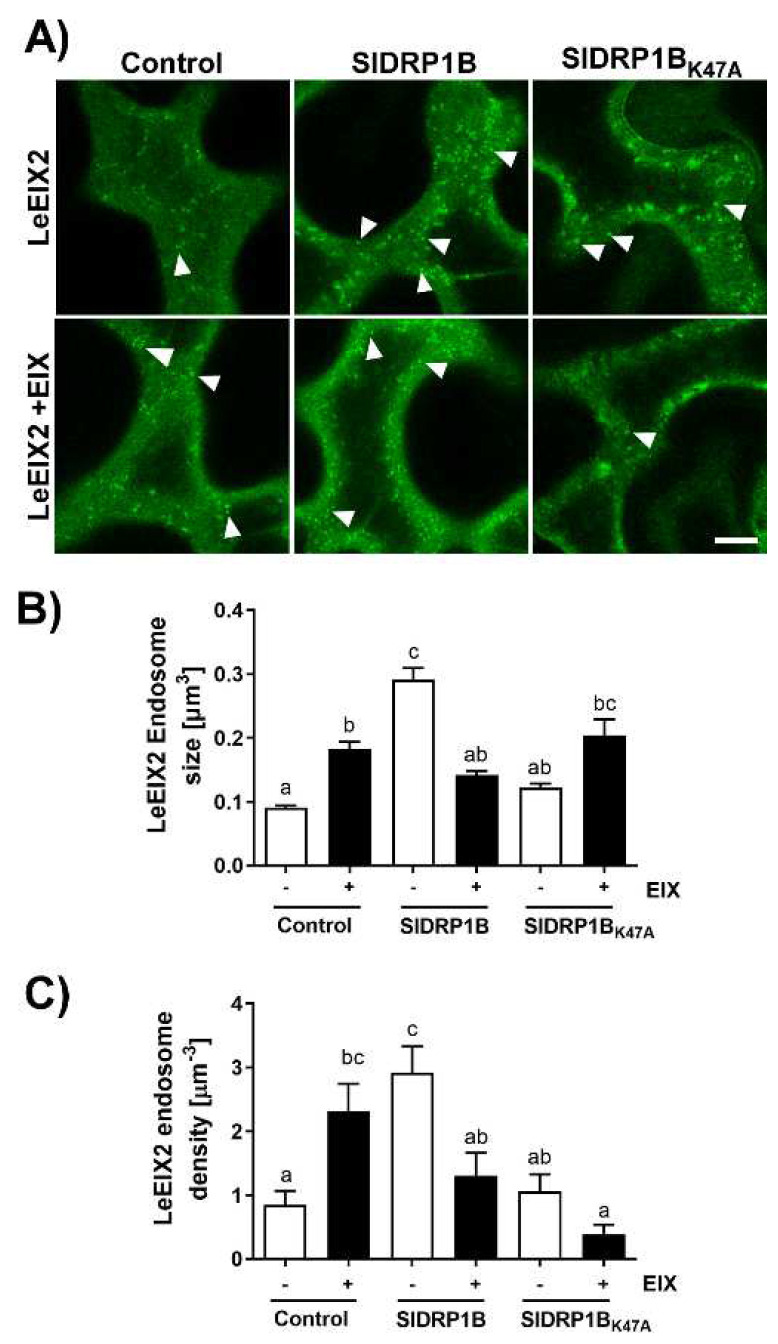
Effect of overexpression of SlDRP1B on LeEIX2 endosomes. *N. benthamiana* leaves transiently expressing LeEIX2-GFP and free mCherry (control), SlDRP1B-mCherry, and SlDRP1B_K47A_, as indicated, were treated with EIX (1 μg mL^−1^ tissue) or water (mock) at the petiole 40 h after transformation and visualized after 15 min. (**A**) Images were acquired using Zeiss LSM 780 confocal microscope (40× objective); arrowheads indicate LeEIX2 endosomes. Scale bar 10 µm. (**B**,**C**) Size and density of LeEIX2 endosomes were quantified using 3D Object Counter (Fiji-ImageJ). LeEIX2 endosomes density was normalized to the control average (mock). Four independent replicates of four images were measured (N = 16). Different letters indicate statistically significant differences between samples in a one-way ANOVA, with Tukey’s multiple comparison test; *p* < 0.05.

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
