# Peer review of "Dynamin-Related Proteins Enhance Tomato Immunity by Mediating Pattern Recognition Receptor Trafficking"

_membranes, 2022, doi:10.3390/membranes12080760_

Round 1

Reviewer 1 Report

Please find the comments in the attached file.

Author Response

We thank you for all the comments and changes suggested, they improve the quality of our manuscript. Here we add a detailed reply for addressing the reviewer's comments:

1.- We kindly ask for an indication of the section that needs to be modified or the software used to detect Plagiarism. We tested in a few online software, and no relevant plagiarism was detected. Therefore, we will gladly make the changes.

2-3.- Corrections were made.

4.- Do Agrobacterium a potential pathogen for tomato? Why authors have used it? Agrobacterium is usually used for transformation in plant and if it act as a pathogen, it would not have been used in plant transformation. Reviewer feel, it is not a suitable pathogen to understand the said purpose.

In nature and fields, Agrobacterium is a pathogen. However, for years plant molecular biologists have used modified Agrobacterium strains for transient and stable transformation. These strains lack virulence genes, have low or no pathogenisity, nut mantain their capacity to transfer DNA from their Ti gene to the plant of interest [1,2].

5.- Suggested to italicize the genus and species names. It was corrected.

6.- Fig 1B: authors are suggested to conduct a qRT-PCR study to validate the fold change value found through geneinvestigator.

Although it would be interesting to corroborate bio-informatic analysis by qRT-PCR, we used this approach to select the best candidate for a functional DRP1. It is not intended to use this approach to characterize the difference between DRP1A and DRP1B. For this and considering that making this analysis will need a longer time to design the strategy and samples, therefore we won't be able to perform the experiment for the current manuscript.

7.- The expression of Solanum DRP1 gene is associated with which treatment?

According to genevestigator analysis, there is a tend to link SlDRP1A increase in expression in treatment related to elicitation and infection, such as EIX (Sl00015), P. infenstans (Sl00022) and T. urticae (Sl00029). On the other hand, treatments related with abiotic stresses do not induce an increase in SlDPR1A gene (Sl00027, Sl00030).

8.- Scale bar was added to each microscope panel

9.-  Author need to mention about over expression in the method section. Also, need to mention about the materials and method for C2H4 production.

Ethylene (C2H4) production methods are described on lines 162-168. We made a change in the legend of figure 4, clarifying that Ethylene = C2H4.

10. Line 327: It was corrected

11. Please check difference between Supplementary Figure and additional Figure. It was corrected.

Reviewer 2 Report

The manuscript of Leibman-Markus et al. deals with the detailed characterization of a member of the DRP family of trafficking regulators from the tomato proteome. Following the pipeline from their previous work focused on SlDRP2A, the authors have now addressed the functional role of SlDRP1B. By generating fluorescent marker lines as well as dominant negative and overexpression SlDRP1B lines, Leibman-Markus et al. analyzed the subcellular localization of the endocytosis regulator, its physical interaction with the RLP LeEIX2, and the significance of this interaction for the LeEIX2 signaling outputs. Besides, a thorough bioinformatics study has elucidated the interactome network of SlDRP1B and the GO clustering of genes co-expressed with SlDRP1B. The described experiments have been carried precisely by including proper controls. Overall, the significance of this study for the plant cell biology field is evident as it contributes to a better understanding of the interplay between ligand-induced endocytosis and immune responses in plant cells. In my opinion, the manuscript deserves publication in Membranes, but only in case the following points are taken into account:

1. In section 3.2 from Results, the authors describe the PM localization of SlDRP1B (lines 242-243). However, the confocal pictures presented in Fig. 2A do not unambiguously demonstrate the nature of the “discrete foci” as the authors do not show co-localization with a PM fluorescent marker. Further experimental evidences have to be provided to support that claim.

2. Quantification of the PM/intracellular fluorescence ratio of LeEIX2-GFP in the loss-of-function mutant and the overexpression SlDRP1B line would be very informative about the RLP pools competent for signaling when combined with the signaling output data (ROS burst and ethylene production). I believe that would be important for testing the validity of the claim for signal propagation from endomembrane compartments.

Minor points:

- Abstract, line 27: SlDRP1A should read as SlDRP2A.

- Results, line 305: “SlDRP1B enhanced” should be changed to “Overexpression of SlDRP1B enhanced”. The same comment for the legend of Fig. 5 describing SlDRP1B-mCherry (line 343) which is also OE.

Author Response

We kindly thank your comments. We believe the manuscript is strengthened, after performing the analysis suggested 

To address them, We performed an colocalization analysis using the plasma membrane marker Flot1 fused to mCherry and SlDRP1B-GFP. These results were added to figure 2 and to the text. In addition, we calculated the ratio of Plasma membrane fluorescence and internal fluorescence of LeEIX2 when Free-mCherry, SlDRP1B-mCherry or SlDRP1BK47A-mCherry is over-expressed. This result was added as supplementary Figure S5. https://drive.google.com/file/d/14S2Nr8OJja4saq40Hkxu3UAv53ZkZVPw/view?usp=sharing

In addition, the minor changes suggested were corrected. 

Round 2

Reviewer 1 Report

Dear authors,

Thanks for your revision. You did not accept to conduct qRT-PCR to validate your experiment. Therefore, the study does not fit good to be get published. I can not understand how it will be too time taking. It can be done in 15 day. By the time you order the primers, you can extract the RNA and prepare the cDNA. I cannot support such invalid experiment.

Regards

Author Response

Dear reviewer,

We understand your point.

And we consider it is possible to have the analysis within 15 days. After that we will send an updated version of the manuscript.

Best regards.